# Diagnostic and Prognostic Value of IL-10, FABP2 and LPS Levels in HCC Patients

**DOI:** 10.3390/medicina59122191

**Published:** 2023-12-17

**Authors:** Egidijus Morkunas, Evelina Vaitkeviciute, Greta Varkalaite, Vidas Pilvinis, Jurgita Skieceviciene, Juozas Kupcinskas

**Affiliations:** 1Department of Gastroenterology, Medical Academy, Lithuanian University of Health Sciences, 50161 Kaunas, Lithuania; juozas.kupcinskas@lsmuni.lt; 2Institute for Digestive Research, Medical Academy, Lithuanian University of Health Sciences, 50161 Kaunas, Lithuania; evelina.vaitkeviciute1@gmail.com (E.V.);; 3Department of Anesthesiology, Medical Academy, Lithuanian University of Health Sciences, 50161 Kaunas, Lithuania

**Keywords:** hepatocellular carcinoma, HCC, prognosis, biomarker, FABP2, IL-10, LPS

## Abstract

Hepatocellular carcinoma (HCC) still lacks valuable diagnostic and prognostic tools. This study aimed to investigate the potential diagnostic and prognostic value of baseline interleukin (IL)-10, fatty acid-binding protein 2 (FABP2) and lipopolysaccharide (LPS) levels in patients with HCC. Serum levels of IL-10, FABP2 and LPS in 47 newly diagnosed HCC patients and 50 healthy individuals were estimated and compared. The best cut-off points for baseline IL-10, FABP2 and LPS levels predicting overall survival (OS) were evaluated. Both levels of FABP2 and IL-10 were significantly higher in HCC patients vs. control group (median 2095 vs. 1772 pg/mL, *p* = 0.026; 9.94 vs. 4.89 pg/mL, *p* < 0.001) and may serve as potential biomarkers in complex HCC diagnostic tools. The cut-off value of 2479 pg/mL for FABP2 was determined to have the highest sensitivity (66.7%) and specificity (55.6%) to distinguish patients with a median OS longer than 17 months. However, the median OS of patients with high and low levels of FABP2 were not significantly different (*p* = 0.896). The prognostic value of LPS as well as FABP2 and IL-10 for HCC patients appears to be limited.

## 1. Introduction

Hepatocellular carcinoma (HCC) is the most common primary liver cancer which is the third leading cause of cancer-associated mortality [1]. In up to 90% of patients HCC develops in a cirrhotic liver and is associated with chronic liver inflammation mostly due to alcoholic liver disease, chronic hepatitis B or C infections and non-alcoholic steatohepatitis [2,3]. Hepatic carcinogenesis is multifactorial process. It is promoted by tumor microenvironment with its local proinflammatory and profibrotic elements as well as systemic inflammatory factors. Several circulating proinflammatory cytokines and chemokines, including interleukin-1α (IL-1α), IL-1β, IL-6, IL-8, and tumor necrosis factor-α (TNF-α), participate in mechanisms of chronic hepatic inflammation and neoplastic transformation of hepatocytes [4,5]. Endotoxins such as lipopolysaccharide (LPS) are not directly involved in chronic inflammation by inducing Kupffer cells to release radical oxygen species (ROS) and proinflammatory cytokines and chemokines [6]. Not only proinflammatory but also anti-inflammatory cytokines, such as IL-10, take part in liver cirrhosis and possibly in HCC development and progression [7]. Microbial translocation from the gut is as well involved in the pathogenesis of liver damage. One of possible markers for increased gut permeability is fatty acid-binding protein 2 (FABP2) [8].

Recent findings show that inflammatory cytokines IL-6 and IL-8 as well as IL-10 are associated with poorer outcomes in HCC patients [7,9]. Loosen and colleagues demonstrated that IL-6 and IL-8 can be valuable in predicting treatment response and survival after transarterial chemoembolization in patients with primary and metastatic liver tumors [9]. A study by Seidensticker et al. evaluated multiple cytokines and delivered cut-off values for IL-6 and IL-8 which were associated with overall survival after 90Y radioembolization in patients with HCC or metastatic disease [10]. The post hoc analysis of the palliative arm of the SORAMIC trial demonstrated that high baseline IL-6 and IL-8 were associated with significantly shorter overall survival in HCC patients undergoing radioembolization or treated with sorafenib monotherapy. Baseline IL-6 and IL-8 with respective cut-off values predicted objective response rates for sorafenib-treated patients [11,12].

As there is evidence that inflammatory molecules such as IL-6 and IL-8 could have diagnostic, prognostic and predictive value for HCC patients, it is important to identify other molecules related to chronic liver inflammation and potentially carcinogenesis which could be involved in future complex tools for better approach for HCC patients.

This exploratory analysis aimed to explore the potential diagnostic and prognostic value of baseline IL-10, LPS and FABP2 in patients with HCC, and provide additional information for possible future complex tools for better HCC diagnosis and prognosis.

## 2. Materials and Methods

The study included 47 patients with newly diagnosed HCC and 50 healthy individuals as controls. Demographic and clinical characteristics of the patients with HCC are provided in Table 1. All participants were enrolled retrospectively at the Department of Gastroenterology of Lithuanian University of Health Sciences from June 2010 to May 2021. Demographic data and clinical parameters were collected at the time of inclusion to the study. Healthy control individuals (volunteers) were free of any chronic diseases and had not received any medications during the previous 3 months prior to inclusion to the study. A schematic of the workflow of the study is shown in Figure 1.

Peripheral blood samples were drawn from all subjects at the time of enrollment in the study. Within 1 h after drawing, the serum samples were placed at −80 °C and stored until further processing. A Human FABP2/I-FABP Quantikine ELISA Kit (DFBP20; R&D Systems, Minneapolis, MN, USA), Human IL-10 Quantikine ELISA Kit (D1000B; R&D Systems, Minneapolis, MN, USA), and Human LPS ELISA Kit (CSB-E09945h; Cusabio, Houston, TX, USA) were used to quantify serum levels of FABP2, IL-10, and LPS in HCC patients and the healthy control subjects.

All statistical analyses were performed using IBM SPSS Statistics 27.0 (Armonk, NY, USA). Numerical data are presented as mean with standard deviation for age and as medians with minimum and maximum values for cytokine levels due to abnormal distribution. For categorical data, results are given as absolute numbers with percentages. For comparison of categorical data, chi-square tests were applied. *t*-tests and Mann–Whitney U tests were used for testing homogeneity and comparing of independent samples in continuous data. ROC curves were used to determine the cut-off value for differences in concentrations of FABP2, IL-10, and LPS that could produce the highest sensitivity and specificity to predict individual survival shorter than the median overall survival. The Kaplan–Meier method was used for estimates of overall survival, and the log-rank test was used to compare survival groups. All tests were carried out two-sided. The level of significance was set to <0.05 without adjusting for multiplicity.

## 3. Results

Blood levels of three possible biomarkers—FABP2, IL-10, and LPS—were measured in both HCC and control groups. Levels of FABP2 and IL-10 were significantly higher in HCC patients vs. control group (median 2345 vs. 1327 pg/mL, *p* = 0.026; 9.94 vs. 4.89 pg/mL, *p* < 0.001). Levels of LPS did not reach significant difference and were higher in healthy patients than in HCC patients (51.95 pg/mL in HCC patients vs. 56.38 pg/mL in control group, *p* = 0.263) (Table 1).

At data lock point for this study, 33 HCC patients (70.2%) had died. The median OS of these patients was 17 months.

Using receiver operating characteristic (ROC) curve analysis, a cut-off value of 2479 pg/mL for FABP2 was determined to have the highest sensitivity (66.7%) and specificity (55.6%) to distinguish patients with a median OS longer than 17 months with the area under the curve (AUC) of 0.622 (Figure 2a). However, the median OS of patients with high and low levels of FABP2 were not significantly different (*p* = 0.893) (Figure 3).

ROC curve analysis could not identify cut-off values for IL-10 and LPS to distinguish HCC patients with shorter and longer median OS than 17 months as AUC levels were too low (0.391 and 0.363, respectively) (Figure 2b,c).

Separate ROC curve analysis was performed discriminating values of FABP2, Il-10 and LPS according to OS of 6 months as well as of 1 and 2 years (Figure 4). The highest AUC (0.622), sensitivity (52.9%) and specificity (84.6%) were obtained for FABP2 to distinguish HCC patients with OS longer than 1 year with the same potential cut-off value of 2479 pg/mL.

## 4. Discussion

In this study, we aimed to investigate LPS, FABP2 and IL-10 as possible diagnostic and prognostic biomarkers in patients with HCC.

HCC development in the chronically inflamed liver tissue is a multifactorial process corresponding to the adaptive immune response and the effector molecules including chemokines, growth factors, metalloproteases, and cytokines. The best studied of them, IL-6 and IL-8 are produced by the tumor-associated macrophages (TAMs) and promote HCC development through different signaling pathways [13,14]. TAMs also suppress the adaptive immune system through expression of high levels of programmed cell death-ligand 1 (PD-L1), causing suppression of the antitumor cytotoxic T-cell responses and stimulation of the tumor growth [15]. Tumor cells themselves produce a variety of inflammatory factors and chemokines to recruit TAMs including IL-6, IL-8, and IL-34 [16]. Recent studies have shown that IL-6 and IL-8 may serve as valuable predictive and prognostic biomarkers in HCC patients. Cut off serum levels distinguishing better and worse survivals than median OS as well as predicting response to various HCC treatment methods were estimated for both IL-6 and IL-8 [9,10,11,12].

Altered gut microbiota composition are associated with chronic liver inflammation and may play a contributory role in hepatic carcinogenesis [17]. Dysbiotic gut microbiota composition affect the gut barrier and increase intestinal permeability, promoting the translocation of gut bacteria [18]. This leads to increased hepatic exposure with gut-derived microbiota-associated molecular patterns that include LPS, a cell wall component of Gram-negative bacteria [19]. Ni et al. demonstrated that LPS-producing genera, such as *Bacteroidetes*, *Firmicutes* and *Fusobacteria,* were increased, while butyrate-producing genera were decreased in early HCC [20]. The accumulation of LPS itself may lead to bacterial translocation and increased gut permeability. Bacterial compounds reach the liver via portal vein and are recognized by pattern recognition receptors, expressed in Kupffer cells, and eliminated under normal conditions [21].

Increasing levels of LPS activate Toll-like receptor 4 signaling pathways, produce proinflammatory cytokines, such as IL-17, TNF-α, IL-6, and IL-1β, and promote liver inflammation. Toll-like receptors also induce tumor proliferation mediated by mitogens such as hepatocyte growth factors, amphiregulin and epiregulin [22]. High concentrations of proinflammatory cytokines producing TAMs are associated with poor HCC prognosis [23]. Plasma LPS concentrations correlate with the degree of liver dysfunction [24].

In this study we found no significant difference in LPS serum levels in healthy individuals vs. HCC patients. On the contrary, the LPS levels were a bit higher in control group. Moreover, this study does not support the measurement of LPS levels for prognostic purposes for HCC patients as the baseline levels do not have impact on the time of survival. It is very likely that LPS concentrations in the liver or the portal system has a higher significance for development and progression of HCC than blood LPS levels.

Increased intestinal permeability may be the primary and one of the most important factors in pathogenesis of chronic liver inflammation by causing bacterial translocation. Enterocytes express FABPs which are thought to be involved in uptake of lipids in the intestine [25]. FABP2, also known as intestinal-type FABP, is rapidly released into the systemic circulation on enterocyte damage and has been shown to be a useful biomarker for diagnosing acute intestinal ischemia [26,27].

Recent studies have demonstrated that fecal FABP2 levels are significantly increased in cases of liver cirrhosis and correlate with disease severity. FABP2 concentration in plasma were found to be different in pattern and absolute levels but as well elevated in liver cirrhosis [28,29]. High FABP2 levels is significantly associated with increased mortality from variceal bleeding in patients with liver cirrhosis [30]. No studies investigating the link between FABP2 levels and HCC were published to date.

This current study demonstrated that serum FABP2 levels were significantly higher in HCC patients compared with healthy controls that might be associated with background of liver cirrhosis in most HCC cases. However, our analysis does not support the idea of measuring FABP2 levels for prognostic purposes in newly diagnosed HCC patients. It is likely that FABP2 is too nonspecific for HCC as it represents only a part of intestinal permeability and is not directly involved in carcinogenesis.

Integrity of the immune system plays the important role in physiologically functioning gut-liver axis. The constant low-level exposure to bacterial components in the liver inhibits the activation of immune cells by specific receptors, such as TLRs, to level, called “endotoxin tolerance”, and activates immune suppression via anti-inflammatory cytokines, such as IL-10, transforming growth factor beta (TGFβ), and hepatocyte growth factor [31]. IL-10 is an anti-inflammatory molecule which limits potentially damaging inflammatory response by inhibiting antigen presentation by dendritic cells and inhibiting macrophage activation and infiltration into damaged liver tissue [32]. At the cellular level, IL-10 is thought to act as a posttranscriptional regulatory agent to suppress the messenger RNA (mRNA) promoting the destabilization of inflammatory cytokine mRNA [33]. Furthermore, IL-10 inhibit apoptotic signaling pathways [34].

Being significantly involved in the development of chronic liver inflammation and HCC, IL-10 should be considered as a potential biomarker for liver damage. Recent studies have shown that some IL-10 polymorphisms could act as significant biomarkers of liver cirrhosis or HCC. Higher serum levels of IL-10 were estimated to be associated with higher inflammatory liver disease severity [7,35].

Our analysis confirms the fact that IL-10 serum levels are significantly higher in HCC patients than in healthy individuals. However, according to our study, it appears to be not suitable as prognostic tool for survival of HCC patients. The increase in IL-10 serum levels may be associated with the progression of the inflammatory process of the liver but not with the carcinogenesis and stage of HCC.

This study has some limitations. It includes rather small number of HCC patients. The HCC group was heterogenous as patients of all BCLC stages were included. It lacks more single patient clinical data such as staging, underlying liver function, results of laboratory tests, radiological imaging, histology availability, performance status, co-morbidities, treatment modalities, cause of death, etc. Therefore, subgroup analysis could not be performed. Larger and more complete studies are needed to better describe the value of these molecules.

In conclusion, our study investigated LPS, FABP2 and IL-10 as possible diagnostic and prognostic biomarkers for newly diagnosed HCC patients. We demonstrated that baseline levels of both FABP2 and IL-10 are elevated in HCC patients and may serve as potential biomarkers in complex HCC diagnostic tools. LPS as well as FABP2 and IL-10 appear not to be suitable for prognosis of survival of HCC patients, however, further investigations are needed.

## Figures and Tables

**Figure 1 medicina-59-02191-f001:**
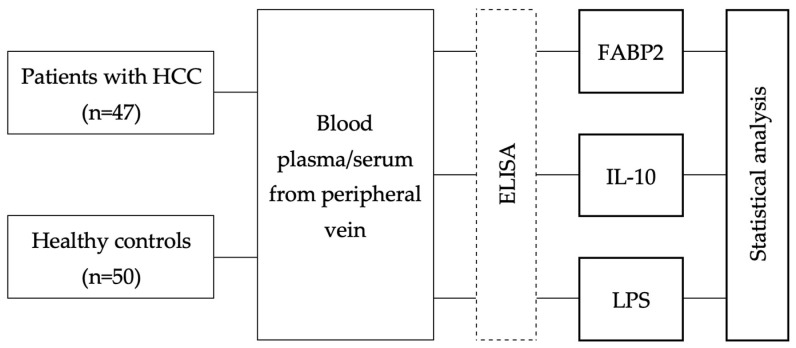
Workflow of the study. HCC; ELISA—enzyme-linked immunosorbent assay; FABP2; IL; LPS.

**Figure 2 medicina-59-02191-f002:**
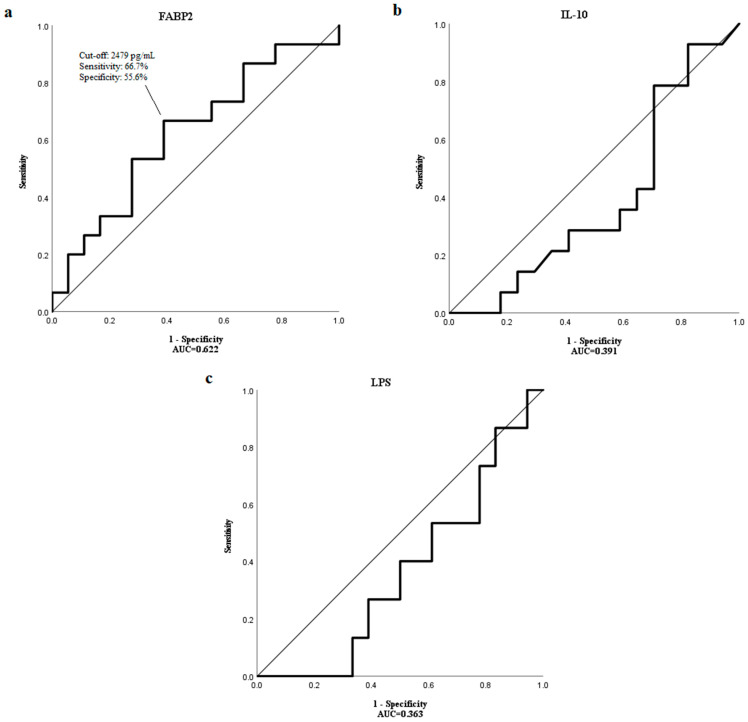
The ROC curve showing the sensitivity and specificity of various cut-off values of baseline (**a**) FABP2, (**b**) IL-10 and (**c**) LPS levels to distinguish patients with the OS longer than a median OS of 17 months.

**Figure 3 medicina-59-02191-f003:**
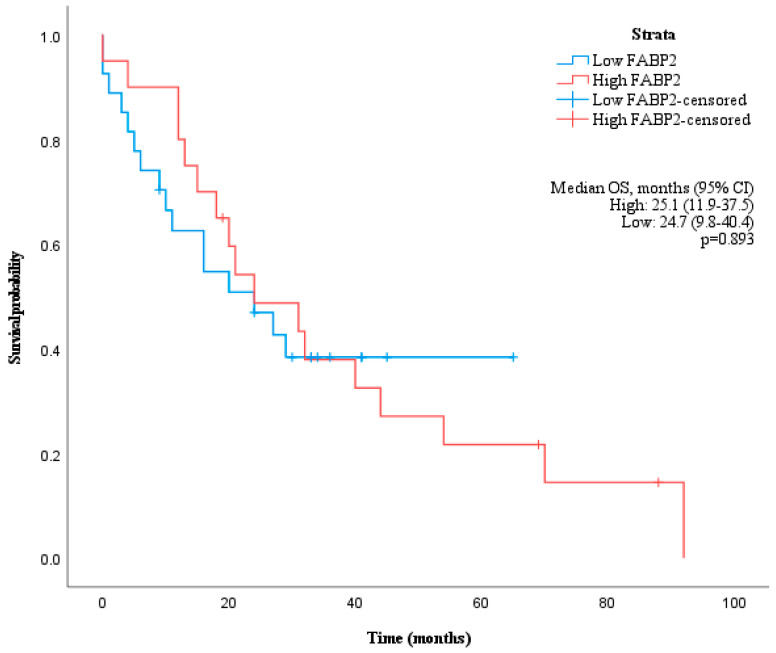
Kaplan–Meier curve showing overall survival of patients grouped by baseline FABP2 values according to cut-off of 2479 pg/mL.

**Figure 4 medicina-59-02191-f004:**
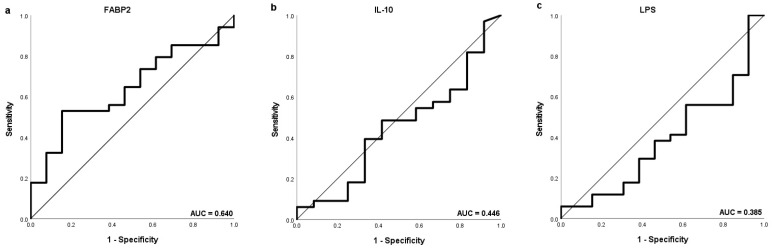
The ROC curve showing the sensitivity and specificity of various cut-off values of baseline (**a**,**d**,**g**) FABP2, (**b**,**e**,**h**) IL-10 and (**c**,**f**,**i**) LPS levels to distinguish patients with the OS longer than 1 year (**a**–**c**), 2 years (**d**–**f**) and 6 months (**g**–**i**).

**Table 1 medicina-59-02191-t001:** Demographic and clinical characteristics of subject groups.

	Controls (n = 50)	HCC Patients (n = 47)	*p* Value
Age, mean ± SD	58.8 ± 4.9	61.3 ± 9.3	0.101
Gender (male)	25 (50%)	39 (83.0%)	<0.001
Liver cirrhosis (yes)		45 (95.7%)	
HCC etiology			
Hepatitis B		7 (14.9%)	
Hepatitis C		21 (44.7%)	
Other (alcohol)		19 (40.4%)	
BCLC stage			
0		2 (4.3%)	
A		15 (31.9%)	
B		17 (36.2%)	
C		7 (14.9%)	
D		6 (12.8%)	
Cytokine levels, median (min-max) (pg/mL)			
FABP2	1327 (518.5–8388)	2345 (326.3–4587)	0.026
IL-10	4.89 (0.036–53) *	9.94 (0.04–564.1) *	<0.001
LPS	56.38 (4.74–436.4)	51.95 (12.37–148.8)	0.263

* Due to insufficient amount of blood for analysis of IL-10, samples of 45 HCC patients and 45 controls were used. SD—standard deviation; HCC—hepatocellular carcinoma; BCLC—the Barcelona Clinic Liver Cancer staging system; min—minimum value; max—maximum value; FABP2—fatty acid-binding protein 2; IL—interleukin; LPS—lipopolysaccharides.

## Data Availability

Data are contained within the article.

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
