# Peer review of "Diagnostic and Prognostic Value of IL-10, FABP2 and LPS Levels in HCC Patients"

_medicina, 2023, doi:10.3390/medicina59122191_

Round 1

Reviewer 1 Report (New Reviewer)

Comments and Suggestions for Authors

Author Response

Reviewer 2 Report (New Reviewer)

Comments and Suggestions for Authors

Thank you for the opportunity to review your manuscript.  The authors present data on potential serum biomarkers on 47 HCC patients with a range of disease progression to 50 healthy controls.  Serum IL-10, FABP2 and LPS levels are investigated for potential predictive value of HCC survival.  

In this study, these markers did not demonstrate useful clinical predictive value.  However, the study is limited in size and in lack of data to better compare groups of varying disease progression.  Overall health of the patients in terms of co-morbidities, treatment attempts, degree of underlying liver function and cause of death are not available.  Though power calculations are not performed, the study is likely underpowered to answer the question of whether these biomarkers are predictive of survival.  Perhaps in combination with other predictive biomarkers, FABP2 could add sensitivity and specificity to predictive models?  Could a larger study that includes tumor type, progression, invasion, metastasis data could better parse out a predictive value for these biomarkers?  As it stands, the manuscript provides data analysis, but is insufficient to properly address the hypothesis of whether these biomarkers have a predictive value in HCC survival. 

Round 2

Reviewer 1 Report (New Reviewer)

Comments and Suggestions for Authors

Author Response

Reviewer 2 Report (New Reviewer)

Comments and Suggestions for Authors

Revision reviewed and the authors have addressed concerns.  Due to inherent limitations, the study provides only a small advance in the field of biomarkers.  However, the research has merit and is well-presented so I feel that the paper is acceptable for publication.  Thank you for involving me in this review. 

Author Response

This manuscript is a resubmission of an earlier submission. The following is a list of the peer review reports and author responses from that submission.

Round 1

Reviewer 1 Report

Comments and Suggestions for Authors

Authors attempted to investigate diagnostic and prognostic values of 3 circulating markers – IL-10, FABP2, LPS in HCC patients.

I do not understand the motivation to investigate those molecules, as the introduction predominantly focuses on IL-8 and IL-6. Also, authors discuss prognostic value of those molecules, not diagnostic tests available for HCC.

Do authors speculate that the liver-gut axis is important in HCC development? This would go towards MAFLD investigation. How many patients had MAFLD in the cohort?

I think overall, the study is underpowered, especially considering heterogeneity in the HCC patient group. Authors should consider including more patients or partnering with another hospital.

Is this a prospective study or were samples obtained from a biobank?

The sensitivity and specificity of FABP2 is very low and I think an underpowered analysis contributes to it.

What is the time frame for HCC diagnosis? Authors state “newly”. Can authors provide details on HCC staging?

More information is needed on healthy controls. Are those patients with benign liver diseases? If they were at the hospital for a specific condition, it should be stated. If they were healthy volunteers, how were they recruited?

How was HCC diagnosed?

I think the rationale for this exploration needs to be explained in more detail.

Subgroup analyses may be helpful to elucidate prognostic values, based on stages, curative surgery, etc.

Overall, I think the topic is of interest and should be pursuit, but I think the study is in its infancy and needs to be continues further before results are presented to the scientific community.

Reviewer 2 Report

Comments and Suggestions for Authors

This study aimed to evaluate three potential biomarkers' diagnostic and prognostic performance. The key findings were increased in FABP2 and IL-10 in patients with HCC. Although it is interesting, several concerns need to be clarified.

Major:

1.     Do you have any rationale for using the cutoff point at 17-month survival to calculate the AUROC of the biomarkers? Could you re-analysis according to the time, such as six months, 1-yr, or 2-yr?

2.     The study's objectives were to evaluate the diagnostic performance of the three biomarkers and prognosis. However, there was no result of diagnosis performance. The authors should clarify this point.

3.     The discussion section's second, third, and fourth paragraphs did not become relevant to the study's findings. Please modify them.

Minor:

1.     Could you please check the wrong words, line 28, "alcoholic liver disease," line 149, "endotoxin tolerance"?

2.     There was missing one participant in HCC etiology. Please check table 1.

3.     Please add SD in LPS values in table 1.